# The Effectiveness of Day Hospitals in the Personal Recovery of Mental Disorder Patients during the COVID-19 Pandemic

**DOI:** 10.3390/healthcare11030413

**Published:** 2023-01-31

**Authors:** Antonio José Sánchez-Guarnido, María Isabel Ruiz-Granados, Javier Herruzo-Cabrera, Carlos Herruzo-Pino

**Affiliations:** 1Hospital Santa Ana, 18600 Motril, Spain; 2Facultad de Ciencias de la Educación, Universidad de Córdoba, 14071 Córdoba, Spain

**Keywords:** mental health, personal recovery, day hospital

## Abstract

Background: In recent years, a new recovery model has gained ground in which recovery is understood as a process of change where individuals are able to improve their health and wellbeing, lead self-sufficient lives and strive to achieve their maximum potential (personal recovery). Despite the existence of data regarding the effectiveness of mental health day hospitals (MHDHs) in reducing relapses in terms of hospital admissions and emergencies, no studies have to date assessed how this change affected the new personal recovery model. Objectives: To verify the effectiveness of MHDHs in improving personal recovery processes among people with mental disorders (MDs). Methods: A prospective cohort study. A group of patients receiving follow-up at MHDHs was compared with another group of patients receiving follow-up in other therapeutic units over a period of three months. Results: Patient recovery at the MHDHs, assessed using the Individual Recovery Outcomes Counter (I.ROC), was found to be significantly better than that of patients attended in other units. Conclusions: MHDHs can contribute to the recovery of people with MDs. This is particularly important at a time when some patients may have experienced impediments to their recovery processes due to the pandemic.

## 1. Introduction

A number of different definitions of mental health day hospitals (MHDHs) have been proposed, and the concept continues to be the subject of debate. In general, they can be understood as partial hospitalization units which provide treatment over limited periods of time, offering structured, coordinated, therapeutically intensive clinical services in a stable therapeutic environment by means of integrated, global schemes that complement recognized approaches to psychological and psychiatric treatment [1,2]. Beyond that, MHDHs can vary, some being conceived as alternatives to full hospitalization and others as alternatives to out-patient treatment. They can also differ with regard to the disorders they handle, with some focusing on psychosis and others on affective disorders, personality disorders, or patients with heterogeneous diagnoses [3,4].

Studies that have examined MHDHs as an alternative to full hospitalization have found them to be just as efficient as full hospitalization [3,5] and even to produce better results in cost efficiency [6,7]. Less evidence is forthcoming about MHDHs created as an alternative to outpatient treatment, although treatment in these MHDHs does seem to result in higher levels of satisfaction and social integration among patients [8], and those units focusing on borderline personality disorders have shown lower numbers of hospitalization cases, stable clinical improvement, better relational functioning and high levels of satisfaction with treatment [6,9,10,11,12,13,14]. Similar results can be seen in MHDHs serving patients with psychotic disorders or heterogeneous diagnoses [7,15].

Although the diversity of programs implemented under the auspices of MHDHs makes it difficult to generalize evidence, the efficiency of such units can nevertheless be said to have been demonstrated in the clinical improvement and stability of their patients [3,16] and in the consequent decrease in relapses in the form of lower numbers of hospital readmissions and visits to emergency services [16,17,18]. The therapeutic objectives of many MHDHs, however, are not only to merely reduce relapse rates but also to achieve full personal recovery within what is known as a recovery model.

In this regard, personal recovery can be defined as a way of living a satisfying, hopeful, contributing life involving the development of new meaning and purpose beyond the effects of mental illness [19]. Patients thus conceptualize personal recovery as the acquisition of control over their own lives, which does not necessary mean returning to a premorbid functional status. This personal recovery model is being used increasingly in MHDHs, with users receiving support to create their own personal models, establish their own objectives, make progress charts and identify strengths and weaknesses [20] MHDHs adhere to a personal recovery model in the sense that they try to facilitate or encourage their patients’ recovery by implementing the model’s practices [21].

Reducing symptoms and supporting recovery are not the same thing, because many people can experience personal recovery even while symptoms of mental illness persist [22]. The distinction between the two is supported by epidemiological research, which suggests that mental health and mental illness can in fact coexist [23]. Other parameters therefore need to be used to evaluate the changes that take place in MHDHs.

It is also true that MD patients suffered particularly during the COVID-19 pandemic, especially during the lockdown periods. MHDHs can play an important role in aiding the recovery of such patients in the aftermath of those periods. COVID-19 clearly had a major impact on the mental health of the general population [24]; [25] especially in people who had previously been diagnosed with MD [26]. We know, for example, that anxiety levels rose more in the MD population than in people with no psychopathological diagnoses [27]. Social distancing also had a particularly negative effect, impeding the social functioning of people with psychotic disorders or other types of MD. As a result, in many cases, recovery processes were delayed, with social, employment, training, leisure and other needs going unmet [28,29].

Paradoxically, during the lockdown, lengths of stay in inpatient units were reduced just when they were most needed [30] and face-to-face interventions at MHDHs were also reduced [31]. This reduction in services is known to have coincided with an increase in relapses and a worsening of patients with psychosis, affective disorders, and personality disorders [32]. In those early stages of the pandemic, MHDHs had to adapt. Remote interventions enabled them to partially alleviate the relapses and hospital admissions but could do little to avoid problems with other aspects of personal recovery [31,33]. Once the most difficult periods of lockdown and social restrictions were over, however, MHDHs were able to resume their crucial role in the recovery process.

Although many MHDHs have sought to base their activity on a personal recovery model, most of the results reported in studies to date have focused on classic indicators such as numbers of relapses, hospital admissions, emergency consultations and symptoms. A study was therefore needed which would include results obtained not only from those classic parameters but also from actual cases of personal recovery once normal face-to-face care in these devices had been recovered, this being a particularly important moment considering what MD patients had gone through during the pandemic.

The objective of this study was therefore to verify the effectiveness of MHDHs in the recovery of people with MDs during the COVID-19 pandemic. The starting hypothesis was that MHDHs contribute to the recovery of such people.

## 2. Materials and Methods

### 2.1. Design

A multicentric prospective cohort study.

### 2.2. Participants

The study included people of both sexes over the age of 18 who had been in follow-up with an MHDH sometime in 2020. The sample size was calculated for a between-group mean difference of 2.5 points in the overall Individual Recovery Outcomes Counter [34] score, a variance of 90, a potential sample loss of 10%, a 95% confidence interval and 90% statistical power. Our collaborators at each MHDH recruited all patients who met the inclusion criteria until the estimated size was reached, the final number being 270 patients from 15 hospitals. All those selected agreed to participate and signed an informed consent form. A total of 244 patients completed the study, so 26 patients who had dropped out of mental health follow-up during those months were lost. The patients were divided into two groups. The first comprised 177 people who had been under treatment at an MHDH from October to December 2020, and the second comprised 67 people who had been treated by other services (43 at community mental health units and 24 at other units) in the same period.

### 2.3. Variables

Having or not having been treated at an MHDH during the period covered by the study (1 October to 31 December, 2020) was used as an independent variable.

The main instrument used as a dependent variable was the Individual Recovery Outcomes Counter [34]. This instrument is self-administered and acts as an overall indicator of an individual’s recovery. It is divided into four domains (Home, Opportunity, People and Empowerment). Home covers aspects associated with having a place to live where one feels safe and protected. Opportunity is a dimension associated with the participant’s possibilities of accessing leisure, education, and employment. People is to do with having friends, people one can trust, and people who can provide support. Empowerment evaluates hope and the extent to which respondents are fully involved in the decisions that affect their lives. I.ROC has a good internal consistency with a Cronbach’s alpha of 0.86 [34,35].

As sociodemographic variables, age, sex, household composition, employment status and level of education were studied.

### 2.4. Procedure

Participants were chosen and their data were collected prospectively by collaborators at each MHDH from the patients’ medical records and through individual interviews. The assessment instrument was administered during an interview with the collaborating professional, in which the patient’s understanding of the content was also checked. One evaluation was carried out in October 2020, and another was carried out three months later in December of the same year. A password-protected database was designed, and the clinical data were processed omitting the patients’ identification information. The study was conducted in accordance with the latest version of the Declaration of Helsinki [36] and following the guidelines of the drug research ethics committees for observational studies at the different hospitals.

### 2.5. Data Analysis

The data were analyzed statistically using IBM SPSS Statistics 21.0 and with a level of statistical significance of *p* < 0.05.

#### 2.5.1. Descriptive Statistics

The results of the categorical variables were expressed as percentages and those of the quantitative variables were expressed as mean, mean differences, standard deviation and Cohen’s d.

#### 2.5.2. Bivariate Analysis

In the between-group analyses, categorical variables were analyzed using the chi-squared test. Quantitative variables were analyzed using Student’s *t*-test for repeated measures and an ANOVA test to identify differences between the groups.

#### 2.5.3. Multivariate Analysis

A multiple linear regression was performed to test the effect of possible contaminating variables.

## 3. Results

### 3.1. Descriptive Analysis of the Sample

The sociodemographic and clinical variables in the two groups are shown in Table 1. With regard to diagnoses, the sample was heterogeneous, with some differences in percentages between the two groups. Nevertheless, in both groups, the most frequent diagnoses were schizophrenia and other psychotic disorders, which were followed by personality disorders and bipolar disorder. The average age of the participants was around 40 years, although it was slightly higher in the group of patients not attending DHs. There were no differences between the groups in sex, household composition or level of education.

#### 3.1.1. Bivariate Analysis

Before and after differences were analyzed within each group (see Table 2), as were differences between the two groups (Table 3). Overall, I.ROC scores in the MHDH group increased significantly (initial M = 39.254; final M = 42.452; t = 4.582; *p* < 0.001). In contrast, changes in the non-MHDH group of patients were not statistically significant (initial M = 43.477; final M = 41.716; t = −1.569; *p* = 0.121). The differences between the scores in the two groups obtained using the ANOVA were found to be significant (d = 0.535; F = 13.947; *p* < 0.001).

Patients being treated at MHDHs improved significantly in the *Home* dimension (initial M: 10.463; final M: 11.463; t = 4.242; *p* < 0.001), while patients not being treated at MHDHs showed no significant changes (initial M = 11.522; final M = 11.238; t = −0.768; *p* = 0.445). Again, the differences between the two groups were statistically significant (d = 0.412; F = 8.273; *p* < 0.004).

In the *Opportunity* dimension, somewhat different results were obtained. As with *Home*, a significant improvement was observed in the MHDH group (initial M = 9.836; final M = 10.468; t = 2.575; *p* = 0.011), but this time, the scores for the control group were found to have worsened significantly (initial M = 10.791; final M= 9.641; t = −2.671; *p* = 0.010). The differences between the groups were also significant (d = 0.52; F = 13.835; *p* < 0.001).

In the *People* dimension, the MHDH group improved significantly (initial M = 9.101; final M = 9.824; t = 3.270; *p* = 0.001), but no significant changes were seen in the group of patients not being treated at MHDHs (initial M = 9.522; final M = 9.656; t = 0.403; *p* = 0.688). In this dimension, the differences between the two groups were not statistically significant (d = 0.20; F = 2.024; *p* < 0.156).

Finally, in the *Empowerment* dimension, a statistically significant improvement was observed in the MHDH group (initial M = 9.853; final M = 10.734; t = 3.718; *p* < 0.001), while scores in the non-MHDH group did not change significantly (initial M = 11.641; final M = 11.179; t = −1.406; *p* = 0.164). Here, the differences between the groups were significant (d = 0.45; F = 9.531; *p* = 0.002).

#### 3.1.2. Multivariate Analysis

A multiple linear regression was performed maintaining the changes in the global I.ROC score during the three months of follow-up as the dependent variable. The independent variable was again the length of stay in MHDHs during the study period. Age and diagnosis, which were the only variables in which there were differences between the two groups, were used as possible contaminating variables. The results show that treatment at MHDHs continued to have a significant effect (B = 5.008; t = 3.593; *p* < 0.001), while no significant association was detected with either age (B = 0.027; t = 0.504; *p* = 0.615) or diagnosis (B = −0.118; t = −0.292; *p* = 0.771).

## 4. Discussion

Our starting hypothesis in this study was that MHDHs can help facilitate personal recovery for people with mental illness. The results obtained confirmed this hypothesis: overall personal recovery scores for people being treated at MHDHs were shown to improve during their treatment, and this improvement was greater than that experienced by people being treated in other units. According to the I.ROC, the test used in the study, this meant that these patients regained hope in their future and in their ability to run their own lives after the toughest part of the COVID-19 pandemic. These results add to the evidence already reported in other studies regarding the efficacy of MHDHs [16,17,37,38,39].

From a general perspective, and considering the overall scores obtained in the I.ROC, patients being treated at MHDHs were found to have made progress in their personal recovery. The results suggest a number of things. Firstly, treatment at MHDHs is able to achieve improvements in personal recovery during the course of the intervention, whereas follow-up at other units does not seem to achieve such progress. Indeed, the non-MHDH patients in our sample were even seen to worsen significantly in one dimension or another (for example, *Opportunity*). The data obtained corroborate the effectiveness of MHDHs as opposed to other services/units [7,15,40,41,42].

Another question raised by the results is that of whether the improvements achieved during the time a patient attends an MHDH persist or are lost over time. Due to the study design, it was not possible to analyze this in depth. We do know, however, that the patients treated at other units during the period of study had attended MHDHs sometime earlier in 2020. Although in certain areas, these patients were found to have experienced no significant deterioration, their scores did fall in the *Opportunity* dimension. In other words, a person’s possibilities of accessing leisure, education and employment decrease when they are not attending an MHDH. This also invites us to consider the need for resources that will safeguard the progress made in a patient’s personal recovery following their discharge from an MHDH and that will provide them with opportunities for employment, training and leisure. People who no longer attend MHDHs would thus still able to keep their hope alive and stay empowered, make the decisions that affect their lives, and retain their life skills, their health habits and their social networks. In practice, however, the possibility of realizing this potential is very much reduced because there are few training, leisure and employment opportunities available to MD patients newly discharged from mental therapy facilities. In our opinion, it is therefore very important for mental health professionals to provide more resources to ensure that patients leaving MHDHs will still have the opportunities to lead a useful life in society through employment, training and leisure activity [43]. This is something we think may currently be receiving insufficient attention in Spain and perhaps also in other countries and which may be impeding potential progress in the personal recovery of people with mental illnesses. Our results concur with data obtained in earlier studies into lack of opportunities for such patients [44,45,46,47].

We believe that the improvement observed in MHDHs in comparison with other therapy units may be related to MHDHs’ ability to offer intensive, integrated, multicomponent treatment for groups, families and individuals: treatment provided by multi-professional teams within an interpersonal setting organized as a therapeutic environment and which takes into account a patient’s ability to lead a satisfactory life guided by their own desires and needs, over and above the constraints imposed by their symptoms [48]. Some studies have already touched on this idea, attributing much of the improvement experienced by patients to MHDHs’ capacity to offer a variety of therapeutic approaches, intensive group experiences and close contact between patients and members of the treatment team [15,49] and also to the coherence and integrity of the ongoing treatment made possible by partial hospitalization [11].

Notwithstanding, the study does have some methodological limitations which need to be taken into consideration. Because it was an observational study, it was not possible to establish cause–effect relationships, and some of the links found may be conditioned due to contamination or interaction with other, non-studied variables (drug use, pharmacological treatment, etc.). The patients in the control group had previously attended MHDHs, making this an atypical control group halfway between a control group and a study group. Furthermore, no details are specified regarding exactly when these patients attended an MHDH or the total duration of their treatment there: when they were studied, they were already receiving follow-up at a variety of different units/services. All these issues must be taken into account when interpreting the results and drawing any conclusions.

Another aspect to be considered is the shortness of the time the patients in the intervention group attended their MHDHs (only 3 months). Some MHDHs work to short timeframes, but interventions in others have much longer durations. These differences are simply part of the heterogeneity intrinsic to MHDHs. Even so, we think that the changes we were attempting to evaluate in this study may possibly have required longer intervention times. For us, the fact that our results showed a clear improvement in recovery within such a short intervention time was something positive. We would expect that over a longer period, these changes would be greater, and we urge other researchers to corroborate this in future studies. It is also important that future studies use a randomized clinical trial design and that patients be appropriately followed up after the end of treatment. Regarding generalization of the results, it is important to bear in mind the specific historical context in which the study was carried out: six months after the start of a pandemic and following a period of lockdown and restrictions which severely affected mental health care resources. It is impossible to know for sure whether the results would have been the same outside a pandemic scenario. Other studies are therefore still needed to verify these results in other contexts and with patients being suitably monitored at the end of their treatment.

In our opinion, these limitations do not detract from the importance of the results. This was a pioneering study in that it was the first to evaluate MHDH results in terms of personal recovery during the COVID-19 pandemic, and the results obtained may inspire future works of research.

## 5. Conclusions

In conclusion, we can say that MHDHs have great potential for aiding people with mental illnesses in their personal recovery processes—a potential greater than that found in other therapy units. They have also facilitated personal recovery processes which could easily have been severely hindered in the wake of the lockdowns and other restrictions imposed during the pandemic. As a final comment, we would also emphasize the need to offer patients opportunities to engage in leisure, training and employment activities on their discharge from MHDHs, so that their personal recovery processes will not come to a halt and they will be able to move on with their lives.

## Figures and Tables

**Table 1 healthcare-11-00413-t001:** Clinical and sociodemographic variables.

	DH Patients N = 177	Other Patients N = 67	*p* Value
Age; mean (SD)	39.01 (11.241)	44.58 (12.157)	F = 11.398 0.001
Sex			X^2^ = 0.72 0.788
Female	97 (54.8%)	38 (56.7%)	
Male	80 (45.2%)	29 (43.3%)	
Household composition			X^2^ = 7.628 0.267
Family of origin	79 (44.7%)	21 (31.3%)	
Own family home (married, living with partner and/or children)	47 (26.6%)	26 (38.8%)	
Horizontal (with friends or siblings)	9 (5.1%)	6 (9%)	
Single person	33 (18.6%)	12 (17.9%)	
Other	9 (5.1%)	2 (3%)	
Employment			X^2^ = 2.590 0.856
Furloughed	1 (0.6%)	1 (1.5)	
Student	13 (7.3%)	3 (4.5%)	
Temporarily unable to work	39 (22.0%)	16 (23.9%)	
Retired, pensioner, or on disability benefit	52 (29.4%)	17 (25.4%)	
Unemployed or not working	51 (28.8%)	23 (34.3%)	
Working	21 (11.9%)	8 (11.9%)	
Level of education			X^2^ = 5.960 0.202
No formal education	3 (1.7%)	3 (4.5%)	
Primary or secondary education	64 (36.2%)	23 (34.3%)	
University entrance level (incl. professional training)	78 (44.1%)	24 (35.8%)	
University-undergraduate	25 (14.1%)	10 (14.9%)	
University-postgraduate	7 (4%)	7 (10%)	
Main diagnosis			X^2^ = 18.179 0.001
1. Schizophrenia and other psychotic disorders	58 (32.8%)	15 (22.4%)	
2. Personality disorders	56 (31.6%)	10 (14.9%)	
3. Bipolar disorder	18 (10.2%)	9 (13.4%)	
4. Depressive disorders	16 (9%)	7 (10.4%)	
5. Other	29 (16.4%) *	26 (38.8%) *	

* Significant differences according to corrected typified residuals.

**Table 2 healthcare-11-00413-t002:** Within-group differences.

	MHDH Group				Non-MHDH Group			
	Initial M (SD)	Final M (SD)	Within-Group Difference (SD)	t (*p*)	Initial M (SD)	Final M (SD)	Within-Group Difference (SD)	t (*p*)
Total I.ROC	39.254 (12.421)	42.452 (12.383)	3.197 (9.284)	4.582 (*p* < 0.001)	43.477 (12.380)	41.716 (12.953)	−1.761 (9.185)	−1.569 (*p* = 0.121)
Home	10.463 (3.597)	11.423 (3.709)	0.960 (3.012)	4.242 (*p* < 0.001)	11.522 (3.59)	11.238 (3.664)	−0.283 (3.024)	−0.768 (*p* = 0.445)
Opportunity	9.836 (3.817)	10.468 (3.729)	0.632 (3.269)	2.575 (*p* = 0.011)	10.79 (3.768)	9.641 (3.365)	−1.149 (3.521)	−2.671 (*p* = 0.010)
People	9.101 (3.447)	9.824 (3.333)	0.723 (2.942)	3.270 (*p* = 0.001)	9.522 (3.483)	9.656 (3.341)	0.134 (2.729)	0.403 (*p* = 0.688)
Empowerment	9.853 (3.964)	10.734 (4.045)	0.881 (3.153)	3.718 (*p* < 0.001)	11.641 (3.922)	11.179 (3.988)	−0.462 (2.693)	−1.406 (*p* = 0.164)

**Table 3 healthcare-11-00413-t003:** Differences between MHDH group and non-MHDH group.

	M Differences	Cohen’s d	Between-Group F	*p*
Total I.ROC	4.958	0.535	13.72	<0.001
Home	1.243	0.412	8.273	0.004
Opportunity	1.781	0.52	13.835	<0.001
People	0.589	0.20	2.024	0.156
Empowerment	1.343	0.45	9.531	0.002

## Data Availability

The data presented in this study are available on request from the corresponding author. The data are not publicly available because they are part of an ongoing project.

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
