# Peer review of "The Effectiveness of Day Hospitals in the Personal Recovery of Mental Disorder Patients during the COVID-19 Pandemic"

_healthcare, 2023, doi:10.3390/healthcare11030413_

Round 1
Reviewer 1 Report
Dear authors, in order to understand the work, you would need a series of clarifications:
1.- What ethics committee authorized the work; and if it is national or regional. Indicate the authorization codes and dates.
2.- Inclusion criteria in the different groups and who was in charge of selecting the patients and in which group were they included?
3.- It is striking that 2 years have passed since the collection of the data, until the sending of this article; Couldn't they update the studio? explain the reasons
4.- The results seem scarce for a work of such magnitude, just three paragraphs; is it a pilot study?
5.- The discussion is extensive compared to the results presented. It would need an in-depth review of this section, providing more up-to-date citations that support or contradict its results.
6.- There are plenty of citations for more than 5 years in the bibliography section. Did you do a web of science search? Update with more recent citations and increase the citations in the discussion section.
Author Response
Dear Reviewer, first of all thank you for your time and review. It has helped us to significantly improve the paper. We have responded to your comments below:
Dear Reviewer, first of all thank you for your time and review. It has helped us to significantly improve the paper. We have responded to your comments below.
1.- Which ethics committee authorized the work; and if it is national or regional. Indicate the authorization codes and dates.
The project was approved by the Granada ethics committee on 07/07/2020 and was valid for the whole of Andalusia, and was also ratified in hospitals outside this community. The resolution of the committee with all the data has been attached to the journal.
2- Inclusion criteria in the different groups and who was in charge of selecting the patients and in which group were they included?
The patients were divided into two groups according to whether or not they were undergoing treatment at an MHDHs during the months of October to December 2020. This is explained in the section on participants. Patient selection was performed by the collaborators of each device. We add this information to the procedure section.
3.- It is striking that 2 years have passed since the collection of the data, until this article was sent; Couldn't they update the study?
The data were collected at a specific moment due to the special interest in seeing the adaptation of the day hospitals after the period of confinement secondary to the Covid-19 pandemic, which meant a setback in the personal recovery processes of the patients. Updating the study at this time would mean carrying out a different study, in another context, somewhat interesting, but with different objectives than those set out in our research. We explain these aspects in the introduction to the article.
4.- The results seem scarce for a work of such magnitude, barely three paragraphs; is it a pilot study?
The study is not a pilot study. An effort has been made to synthesize the main results of the study between tables and the minimum necessary text. Even so, we are grateful for your comments and have decided to expand the section by including descriptive analysis of the groups and a multivariate analysis.
5.- The discussion is extensive in comparison with the results presented. It would need a thorough revision of this section, providing more updated citations that support or contradict its results.
The review is expanded and updated citations are provided.
6.- There is an abundance of citations older than 5 years in the bibliography section. Did you search the Web of Science? Update with more recent citations and increase citations in the discussion section.
We performed a new search in Web of Science and took the opportunity to update the bibliography with more recent citations.
Reviewer 2 Report
It is a valuable article, the following are suggested:
1- In the abstract section, the abbreviation MHDH should be written in full.
2- In the Materials and Methods section, the characteristics of the study participants are written, which should be transferred to the results section. In the method section, only the method of sampling should be mentioned, but the result of sampling is related to the results section.
3- Please explain how the number of samples changed during the three months of the study and how did you deal with those who did not want to participate in the study or who withdrew during the study? Was the consent to participate in the study obtained from the patient or the patient's family? Please explain in the text.
4- The study evaluation tool is a self-administered instrument. How did you make sure that the patient with schizophrenia has a correct understanding of the content of this tool?
5- Based on the results of this study, what research do you suggest for the future? please explain in the discussion section.
Author Response
Dear Reviewer, first of all thank you for your time and review. It has helped us to significantly improve the paper. We have responded to your comments below:
1- In the summary section, the abbreviation MHDH should be written in full.
We agree. We change it.
2- In the Materials and Methods section, the characteristics of the study participants are written, which should be transferred to the results section. In the method section, only the sampling method should be mentioned, but the sampling result is related to the results section.
We agree. We moved it to the results section.
3- Explain how the number of samples changed during the three months of the study and how you dealt with those who did not want to participate in the study or who withdrew during the study. Was consent to participate in the study obtained from the patient or the patient's family? Please explain in the text.
A total of 270 patients met the criteria for participation; all those selected agreed to participate and signed an informed consent form. A total of 244 patients completed the study, so 26 patients who dropped out of the mental health follow-up during those months were lost.
We add this information in the manuscript.
4- The assessment tool of the study is a self-administered instrument. How did you ensure that the patient with schizophrenia correctly understood the content of this tool?
The tool was administered during an interview with the collaborating professional and the collaborating professional checked the patient's understanding of the content.
We add this information in the manuscript.
5- Based on the results of this study, what research do you suggest for the future, please explain in the discussion section.
We encourage other investigators to conduct a randomized clinical trial with longer intervention time and adequate subsequent follow-up.
We specify this suggestion in the discussion
Author Response
Dear Reviewer, first of all thank you for your time and review. It has helped us to significantly improve the paper. We have responded to your comments below:
Mental disorders are one of the greatest challenges facing all countries. They are the second largest cause of disease burden in high- and middle-developed countries, costing the national economy billions of dollars in direct health expenditure and lost productivity.
No model of psychiatric service system is perfect and different countries have different service delivery models. Still, some organizational solutions are more attractive and effective than others.
But Tempora mutantur et nos mutamur de illis. Hence, evaluation research must be conducted to obtain the scientific rationale for evidence-based decision making.
The paper "Efficacy of day hospitals in the recovery of the mentally ill during the COVID-19 pandemic" is a good example. The paper presented by Antonio José Sánchez-Guarnido and his collaborators has ten pages divided into the typical parts: introduction, material and methods, results, discussion and conclusions. It is an excellent work: study and effect. The topic is essential, the research is properly conducted; the methods are pertinent; the presentation is clear and logical, the discussion is correct," the sense of limitations" is complete, and the conclusions are reliable.
But before making the decisive decision to publish, I think it would be worthwhile to discuss some minor issues.
First, the notion of "mental health patients" in the title. I think it would be better to say "patients with mental disorders". Note that those words are in the abstract.
Firstly, the notion of "Mental health patients" in the title. I think it would be better to say, "Mental disorders patients". Notice that such words are in the abstract.
We agree, we change it.
The next question is the notion of "day hospital", which has been used for many years, starting in Canada just after World War II, and is well known and understandable. Don't you think the time has come to discuss and, if necessary, change the formulation? A hospital has a clear definition and it is the hospital. The "Extra Muros" facility is not the hospital and therefore should have another description. It is fine to subject it to a scoping review, no doubt, at some point in the future.
The next issue is the notion of a "day hospital. "It has been used for many years, beginning in Canada just after the second WW was ended, and is well-known and understandable. Do you think, anyway, that it is time to discuss and eventually change the wording? A hospital has a clear-cut definition and is the hospital. "Extra Muros" facilities are not the hospital and, therefore,should have another description. It is fine to issue for scoping review, certainly, somewhere in the future.
The concept of the day hospital is still under continuous debate and it would certainly be very interesting to subject it to a revision of its scope. We add a small commentary on this subject in the introduction.
The first sentence of the introduction has an error /" It is not........./.
The first sentence of the introduction has an error /" The It is not........./.
Thank you. We corrected it.
In the end, why did the document begin with such a pessimistic tone? "It is not easy to find.........'"It would be better to say, for example, some general sentences about the importance of psychiatric day centers.
In the end, why did You start the paper in such a pessimistic mood? "It is not easy to find.........'"It would be better to say, for example, a few general sentences about the importance of psychiatric day facilities.
Ok. We change the beginning of the document.
Reviewer 4 Report
The article "The Effectiveness of Day Hospitals in the Recovery of Mental Health Patients during the COVID-19 Pandemic" aims to verify the effectiveness of mental health day hospitals in the recovery of a Spanish sample with mental disorders during the COVID-19 pandemic
The study is interesting and the manuscript is well written and was pleasant to read. However, I have some comments which are listed below:
Abstract
Avoid using abbreviations in the abstract.
Introduction
- Line 27: Delete “The”.
- Line 72: Since this is the first time you are using the abbreviation “MD”, also write the full word “Mental Disorder”
- On lines 66-67 you rightly point out the difference between clinical and personal recovery (Yu et al. 2022). Since your aim is to discuss personal recovery and to avoid confusion, I would like you to refer to it as "personal recovery" in the title, keywords, and the rest of the manuscript (e.g. line 99, 100, etc.).
Bibliography: Yu, Y., Shen, M., Niu, L., Liu, Y. E., Xiao, S., & Tebes, J. K. (2022). The relationship between clinical recovery and personal recovery among people living with schizophrenia: A serial mediation model and the role of disability and quality of life. Schizophrenia research, 239, 168–175. https://doi.org/10.1016/j.schres.2021.11.043
- Lines 72-82: Explain in more detail and with supporting literature the impact that COVID-19 pandemic have had on the worsening of severe mental disorders such as schizophrenia or bipolar disorder. It would be interesting to compare how the COVID-19 pandemic has changed psychiatric hospitalization rates and diagnostic patterns of severe mental disorders (Peraire et al. 2022), and the role MDHD would have in decreasing these hospitalizations and improving patients' clinical and personal recovery.
Bibliography: Peraire, M., Guinot, C., Villar, M., Benito, A., Echeverria, I., & Haro, G. (2022). Profile changes in admissions to a psychiatric hospitalisation unit over 15 years (2006-2021), considering the impact of the pandemic caused by SARS-CoV-2. Psychiatry research, 320, 115003. https://doi.org/10.1016/j.psychres.2022.115003
Material and methods
- Is the instrument you used (I.ROC) validated in Spanish?
- How did you calculate the necessary sample size? How where the patients recruited?
- Line 111. What were these “other units”? Were hospitalisation units?
- How were patients from MHDH and other services treated: face-to-face or telematically?
- What was the average length of stay of patients in the MHDH and other services before entering the study? Could this length of stay influence the personal recovery?
- Given that drug use is very commonly associated with mental disorders (dual pathology) and is related to worse symptomatic, functional and personal recovery (Peralta et al., 2022), has this aspect been evaluated among MHDH patients and non patients? If not, add as a limitation.
Bibliography: Peralta, V., García de Jalón, E., Moreno-Izco, L., Peralta, D., Janda, L., Sánchez-Torres, A. M., Cuesta, M. J., & SEGPEPs Group (2022). Long-Term Outcomes of First-Admission Psychosis: A Naturalistic 21-Year Follow-Up Study of Symptomatic, Functional and Personal Recovery and Their Baseline Predictors. Schizophrenia bulletin, 48(3), 631–642. https://doi.org/10.1093/schbul/sbab145
- Table 1. Add the Corrected Typified Residuals to find out in which categories there are significant differences.
- Line 134. I assume you meant to write "with a Cronbach's alpha of 0.86"
- Please consider doing some linear and logistic regressions on the possible influence of age and main diagnosis on I.ROC scores. If any, comment on it in the discussion section.
Author Response
Dear Reviewer, first of all thank you for your time and review. It has helped us to significantly improve the paper. We have responded to your comments below:
The article " The efficacy of day hospitals in the recovery of mental health patients during the COVID-19 pandemic " aims to test the efficacy of mental health day hospitals in the recovery of a Spanish sample with mental disorders during the COVID-19 pandemic.
The study is interesting and the manuscript is well written and was enjoyable to read. However, I have a few comments which are listed below:
Summary
Avoid using abbreviations in the abstract.
Introduction
- Line 27: Delete "The".
Ok. Corrected.
- Line 72: Since this is the first time you are using the abbreviation "MD", please also write the full word "Mental Disorder".
Ok. Corrected.
- In lines 66-67 you rightly point out the difference between clinical and personal recovery (Yu et al. 2022). Since your goal is to discuss personal recovery and avoid confusion, I would like you to refer to it as "personal recovery" in the title, keywords, and the rest of the manuscript (e.g., line 99, 100, etc.).
Bibliography: Yu, Y., Shen, M., Niu, L., Liu, YE, Xiao, S. and Tebes, JK (2022). The relationship between clinical recovery and personal recovery among people living with schizophrenia: a serial mediation model and the role of disability and quality of life. Schizophrenia Research, 239, 168-175. https://doi.org/10.1016/j.schres.2021.11.043.
Ok. We think it is adequate. We change it.
- Lines 72-82: Explain in more detail and with supporting literature the impact that the COVID-19 pandemic has had on the worsening of severe mental disorders such as schizophrenia or bipolar disorder. It would be interesting to compare how the COVID-19 pandemic has changed psychiatric hospitalization rates and diagnostic patterns of severe mental disorders (Peraire et al. 2022), and the role that MDHD would have in decreasing these hospitalizations and improving clinical and personal recovery of patients .
Bibliography: Peraire, M., Guinot, C., Villar, M., Benito, A., Echeverría, I., & Haro, G. (2022). Evolution of the profile of admissions to a psychiatric inpatient unit over 15 years (2006-2021), considering the impact of the pandemic caused by SARS-CoV-2. Research in Psychiatry , 320 , 115003. https://doi.org/10.1016/j.psychres.2022.115003.
Ok. We will add comments and references in this regard.
material and methods
- Is the instrument you used (I.ROC) validated in Spanish?
The first author of this article carried out an adaptation and validation of the instrument in Spain.
Jose Antonio Garrido-Cervera, Sánchez-Guarnido, A. J., Huizing, E., & Cuesta-Vargas, A. (2019). Psychometric Properties at Spanish Population of the The Individual Recovery Outcomes Counter (I.ROC).. XII International and XVII National Congress of Clinical Psychology.
- How did you calculate the required sample size? How were patients recruited?
Sample size was calculated for a between-group mean difference of 2.5 points in the overall I.ROC score, a variance of 88, a potential sample loss of 10%, a 95% confidence interval, and 90% statistical power. The collaborators of each MHDH recruited all patients who met the inclusion criteria until the estimated size was reached.
This information is added in the text.
- Line 111. What were these "other units"? Were they inpatient units?
There were no full inpatient units. Most of these units were community mental health units.
Information on these units is also specified in the text.
- How were MHDH patients and other services attended to: face-to-face or telematically?
During the study period, attendance in person had already been recovered. This aspect is also clarified in the text.
- What was the average length of stay of patients in the MHDH and other services before entering the study? Can this length of stay influence personal recovery?
We are not aware of this information. We include it as a limitation of the study in the discussion.
- Given that drug use is very commonly associated with mental disorders (dual pathology) and is related to worse symptomatic, functional and personal recovery (Peralta et al., 2022), has this aspect been assessed among MHDH patients and non-patients? If not, add it as a limitation.
Bibliography: Peralta, V., García de Jalón, E., Moreno-Izco, L., Peralta, D., Janda, L., Sánchez-Torres, AM, Cuesta, MJ, & SEGPEPs Group (2022). Long-term outcomes of first admission psychosis: a 21-year naturalistic follow-up study of symptomatic, functional, and personal recovery and its baseline predictors. Schizophrenia Bulletin, 48(3), 631-642. https://doi.org/10.1093/schbul/sbab145.
This aspect has not been evaluated. We include it as a limitation.
- Table 1. Sum of Corrected Typed Residuals to find out in which categories there are significant differences.
Thank you. We do. We point out in the table the categories with significant differences according to Typified Corrected Residues.
- Line 134. I assume you meant to write "with a Cronbach's alpha of 0.86".
Right, we corrected it.
- Consider running some linear and logistic regressions on the possible influence of age and principal diagnosis on I.ROC scores. If any, comment on it in the discussion section.
We think it is a very good idea. We add the results in the text.
Round 2
Reviewer 1 Report
The changes made increase the scientific quality and respond to the requirements of this reviewer.Reviewer 4 Report
The authors answered all my questions. Thank you.
P.S. Note that in line 87 you wrote (Peraire er al., 2023) instead of (Peraire et al., 2023).